# Optimal-er Auctions through Attention

**Dmitry Ivanov**[*]
HSE University & Technion
Israel

**Iskander Safiulin**
Independent researcher
Russia

**Igor Filippov**
Independent researcher
Russia

**Ksenia Balabaeva**
ITMO University & BIOCAD
Russia

## Abstract

RegretNet is a recent breakthrough in the automated design of revenue-maximizing auctions. It combines the flexibility of deep learning with the regret-based approach to relax the Incentive Compatibility (IC) constraint (that participants prefer to bid truthfully) in order to approximate optimal auctions. We propose two independent improvements of RegretNet. The first is a neural architecture denoted as Regret-Former that is based on attention layers. The second is a loss function that requires explicit specification of an acceptable IC violation denoted as regret budget. We investigate both modifications in an extensive experimental study that includes settings with constant and inconstant numbers of items and participants, as well as novel validation procedures tailored to regret-based approaches. We find that RegretFormer consistently outperforms RegretNet in revenue (i.e. is *optimal-er*) and that our loss function both simplifies hyperparameter tuning and allows to unambiguously control the revenue-regret trade-off by selecting the regret budget.[2]

## 1 Introduction

Several decades ago, Myerson [42] has proposed a general solution for the problem of optimal (revenue-maximizing) design of single-item auctions. Despite the collaborative effort of the community, the results of the same generality for the multi-item auctions have not been obtained. As an alternative to analytical solutions, automated auction design [7, 8] takes a perspective of constrained optimization. In particular, the objective is to maximize the revenue while satisfying the Incentive Compatibility (IC) and Individual Rationality (IR) constraints. This framework can provide approximate solutions in settings where the optimal mechanisms are unknown, as well as hint at what the analytical solutions may look like. Whereas the early algorithms utilize linear programming and classic machine learning, modern algorithms focus on deep learning.

Dütting et al. [17] are the pioneers of this approach with their RegretNet architecture. RegretNet is a neural network that represents a mechanism by mapping a matrix of bids that $n$ participants report for $m$ items into a probabilistic allocation matrix and a payment vector. It is trained via differential optimization on a mixture of two objectives: to maximize revenue and to minimize regret, which is a relaxation of the IC constraint proposed in [15]. RegretNet recovers near-optimal revenue in the analytically solved settings and outperforms the previous state-of-the-art while having vanishing regret guarantees. In this paper, we propose two independent improvements of RegretNet.

First, we propose a neural architecture based on attention layers. We name it RegretFormer after the widely-known Transformers [60]. An illustration of the architecture is provided in Figure 1.

---

[*]dimonenka@mail.ru, divanov@campus.technion.ac.il

[2]Code is available here

36th Conference on Neural Information Processing Systems (NeurIPS 2022).

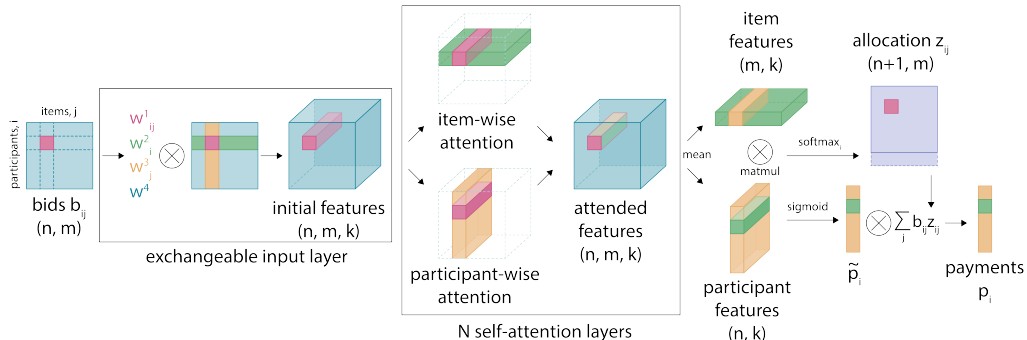

Figure 1: Our architecture RegretFormer. The description is provided in Section 3.1.

RegretFormer has several crucial properties. On the one hand, it is by design insensitive to the order of items and participants, which is desirable for learning symmetric auctions. Note that it can also be modified to learn asymmetric mechanisms. On the other hand, it does not assume a predetermined constant input size, which allows one network to learn from data consisting of auctions of varying sizes, as well as to generalize to settings unseen during training.

Second, we propose an alternative objective. Instead of optimizing arbitrary mixtures of revenue and regret, our approach maximizes revenue given an explicitly specified regret budget, which is an acceptable violation of the IC constraint. This is technically implemented through the method of Lagrange multipliers and dual gradient descent. Our approach allows to unambiguously balance revenue and regret by varying the regret budget. As a bonus, we find that our loss function is significantly less sensitive to the choice of loss-related hyperparameters.

We conduct an extensive experimental study to verify the strengths of our modifications. Specifically, we compare our and existing architectures in both symmetric and asymmetric settings with constant input size, as well as in novel settings with inconstant input size. We find that RegretFormer consistently outperforms existing architectures in revenue given the same regret budget. We additionally verify this result with two novel validation procedures. The first procedure is based on estimating the regret of one network given the optimal misreports approximated by another network. The second procedure is based on network distillation. Regarding our loss modification, we confirm that specifying the regret budget results in the intended revenue-regret trades-off.

## 2 Background

### 2.1 Problem statement

Let $N = \{1, ..., n\}$ be a set of $n$ bidders, $M = \{1, ..., m\}$ be a set of $m$ items. Each bidder $i$ evaluates all items with valuation function $v_i : 2^M \to \mathbb{R}$. In our study, we only consider *additive* valuations, meaning that for each item $j \in M$ a bidder has a valuation function $v_i(j)$, and a valuation of a subset $S \subseteq M$ equals to the sum of valuations of the items in the subset $v_i(S) = \Sigma_{j \in S} v_i(j)$. The valuation function can also be extended to evaluate probabilistic assignments: $v_i(z_{ij}) = z_{ij} v_i(j)$, where $z_{ij}$ is the probability that participant $i$ is assigned item $j$.

The valuation function $v_i$ is drawn independently from the distribution $F_i$ over all possible valuation functions $V_i$. A set of valuations of all bidders forms a valuation profile $v = (v_1, ..., v_n)$. An auctioneer does not know the bidders' valuations nor their distribution $F = (F_1, ..., F_n)$, but has a sample $L = (v^1, ..., v^{|L|})$ of valuation profiles drawn from $F$. The bidders independently report their bids $b = (b_1, ..., b_n) \in V$, based on which the auction allocates the items and charges payments. Formally, the auction is defined as a tuple $(g = (g_1, ..., g_n), p = (p_1, ..., p_n))$ of probabilistic allocation rules $g_i : V \to [0, 1]^M$ and payment rules $p_i : V \to \mathbb{R}_{\geq}$.

Define the bidder's utility as $u_i(v_i, b) = v_i(g_i(b)) - p_i(b)$. Define the valuation profile without the valuation $v_i$ as $v_{-i}$. Similarly, define the bids without $b_i$ as $b_{-i}$, and the valuation profile of all bidders except $i$ as $V_{-i} = \Pi_{k \neq i} V_k$. An auction is *dominant strategy incentive compatible* (DSIC) if the utility

of each bidder is maximized by reporting their true valuations regardless of bids of other bidders: $\forall i, v, b : u_i(v_i, v) \geq u_i(v_i, (b_i, v_{-i}))$. An auction is *ex-post individually rational* (IR) if each agent's utility is non-negative for all possible valuations and bids: $\forall i, v, b : u_i(v_i, (b_i, v_{-i})) \geqslant 0$.

Define the revenue of a profile as a sum of bidders' payments $\Sigma_i p_i(v)$. The goal of the optimal auction design is to maximize the expected revenue subject to DSIC and IR constraints. The problem is analytically solved for the auctions with one item in the seminal work of Myerson [42], provided the valuation distributions are known. There are no general solutions for auctions with $m > 1$.

The task of optimal auction design can be viewed as an optimization or a machine learning problem. In this case, the expected revenue takes the place of the objective. There is a class of auctions $(g(w), p(w)) \in M$ parameterized with $w \in \mathbb{R}^d$ for some $d \in \mathbb{N}$ and a set of bidder valuation profiles $L$ drawn as independent and identically distributed variables from $F$. Then, the goal of the optimization procedure is to find an auction that minimizes the negated expected revenue $-\mathbb{E}_F[\Sigma_{i \in N} p_i(v; w)]$ while satisfying DSIC and IR constraints.

## 2.2 RegretNet

RegretNet [17] is a deep learning solution for the optimal auction design. It can be used for multi-bidder and multi-item settings, the analytical solutions for which are unknown.

The architecture of RegretNet consists of two networks: the allocation network and the payment network. Both networks take flattened constant-sized bid matrix $B^{nm}$ as an input and process it through several fully-connected layers.

The allocation network maps a matrix of bids to categorical distributions of allocations between participants for each item: $A_{net}(B^{nm}) = Z^{nm}$, where $z_{ij}$ is the probability of allocating the $j$-th item to the $i$-th bidder. To allow for the possibility of an item remaining unallocated, the output of the final layer has the size of $(n+1) \cdot m$, imitating an additional dummy participant with zero valuations. Then, to obtain properly scaled allocation probabilities over participants for each item, the softmax activation is applied to the output of the final layer corresponding to each item. Specifically, $z_{ij} = e^{\tilde{z}_{ij}} / \sum_{i=1}^{n+1} e^{\tilde{z}_{ij}}$, where $\tilde{z}_{ij}$ denotes the unnormalized allocation probability (i.e. logit).

The payment network maps a matrix of bids to a vector of pay values: $P_{net}(B^{nm}) = \hat{P}^n$, where $\hat{p}_i$ is the fraction of the $i$-th bidder expected utility that the bidder transfers to the auctioneer. The payment is then computed as $p_i = \hat{p}_i \sum_{j=1}^{m} z_{ij} b_{ij}$ for $i = 1, \ldots, n$. To properly scale $\hat{p}_i \in [0, 1]$, a sigmoid activation is applied to the output of the final layer. Because the payment cannot exceed the bidder's expected utility, the mechanisms discovered by RegretNet always satisfy the IR constraint.

Given a sample of profiles $L$, RegretNet aims to optimize the following empirical objective:

$$\min_w \quad -\frac{1}{|L|} \sum_{l \in L} \sum_{i \in N} p_i(v^l; w)$$
$$s.t. \quad rgt_i(v^l; w) = 0, \quad \forall i \in N, l \in L \tag{1}$$

where $rgt_i$ denotes the $i$-th bidder's ex-post regret. Regret is a quantifiable relaxation of the DSIC constraint first introduced in [15]. It is defined as the difference between the $i$-th bidder's utilities given the optimal misreport (that provides the highest utility) and the truthful bid:

$$rgt_i(v^l; w) = \max_{v_i'} \left( u(v_i^l, (v_i', v_{-i}^l); w) - u(v_i^l, v^l; w) \right) \tag{2}$$

To solve the constrained optimization problem 1 over the space of the network parameters $w$, the authors of RegretNet employ the augmented Lagrangian method. This yields a loss function that combines revenue maximization with a penalty for violations of DSIC constraints:

$$\mathcal{L}_{outer}(w) = \sum_{i \in N} \left[ -P_i + \lambda_i \widetilde{R}_i + \frac{\rho}{2} \widetilde{R}_i^2 \right] \tag{3}$$

where $B$ denotes mini-batch, $P_i = \frac{1}{|B|} \sum_{l \in B} p_i(v^l; w)$ is the average revenue from the $i$-th participant, $\widetilde{R}_i = \frac{1}{|B|} \sum_{l \in B} \widetilde{rgt}_i(v_i'^l, v^l; w)$ is the average approximate regret of the $i$-th participant, and $\widetilde{rgt}_i(v_i'^l, v^l; w) = \big(u(v_i^l, (v_i', v_{-i}^l); w) - u(v_i^l, v^l; w)\big)$ is an approximation of $rgt_i$. To find the approximate regret, the optimal misreports $v_i'^l$ are estimated via gradient descent in the inner optimization loop by minimizing the $i$-th participant negated utility as a function of their misreport:

$$\mathcal{L}_{inner}(v_i'^l) = -u(v_i^l, (v_i'^l, v_{-i}^l); w) \tag{4}$$

The Lagrange multipliers $\lambda_i$ and $\rho$ are periodically updated throughout the training according to the schedules: $\lambda_i \leftarrow \lambda_i + \rho \widetilde{R}_i$ and $\rho \leftarrow \rho + \rho_\Delta$. The starting values of $\lambda_i$ and $\rho$ and the learning rate $\rho_\Delta$ are hyperparameters that control the trade-off between revenue and regret of the learned mechanism.

### 2.3 EquivariantNet

EquivariantNet is an alternative architecture proposed by Rahme et al. [51] to effectively deal with symmetric auctions. It relies on a theorem originally proven by [10] that there exists an optimal solution for a symmetric auction that is permutation-equivariant, i.e. insensitive to the order of inputs.

Like RegretNet, EquivariantNet has an allocation network and a payment network. Both networks consist of compositions of exchangeable layers that preserve permutation-equivariance [25]. Their definition is provided in the Appendix. The authors observe that EquiavariantNet produces competitive results but does not outperform RegretNet in revenue.

### 2.4 Related work

Early work that explores the automated solutions to auction design formulates the problem as linear program [7, 54, 8] or searches within specific families of DSIC auctions [35, 55]. The former approach suffers from scalability issues due to the curse of dimensionality [24], while the latter approach searches within a limited family of auctions that might not contain the optimal mechanism. Classic machine learning has also been applied to the auction design [34, 15, 44], but these approaches are considered less flexible and general than the deep learning alternatives [17].

In recent years, multiple extensions of RegretNet have been proposed, including the extensions to the optimal auction design with budget constraints [18], fairness constraints [33], and human preferences over desirable allocations [49], as well as the problem of faculty allocation [19] and the matching problem [53]. Shen et al. [56] and Dütting et al. [17] propose alternatives to RegretNet that are exactly DSIC but that are only applicable to auctions with one bidder. Curry et al. [9] combine the approach of searching within limited families of DSIC auctions with deep learning. Rahme et al. [52] propose a simplified loss function for RegretNet that is easier to tune, as well as a potentially faster regret estimation procedure based on training an adversarial bidder network. Deep learning has been applied to other aspects of mechanism design, such as iterative combinatorial auctions [62], minimizing economic burden on bidders within the Groves family of auctions [58], optimal bidding strategies [46, 45], optimal redistribution mechanisms [38], and E-commerce advertising [36]. Several studies have combined reinforcement learning with auction design [59, 29, 47, 1]. A plethora of research has also focused on the questions of sample complexity of designing revenue-maximizing auctions [6, 41, 13, 3, 40, 39, 43, 57, 20, 28, 26, 23, 21].

## 3 Our modifications of RegretNet

In this section, we describe our two modifications of RegretNet. The first modification is a novel neural architecture for the optimal auction design, which we denote as RegretFormer. The second modification is an alternative constrained objective that yields a more convenient loss function.

### 3.1 RegretFormer: enhancing RegretNet with attention layers

The neural architecture of RegretNet has several issues. One issue is the sensitivity of RegretNet to the order of items and participants in the input bid matrix. Such order should not affect the outcome of a symmetric auction [10, 50]. Even in non-symmetric auctions, permutation-equivariant

solutions sometimes achieve near-optimal revenue [27, 2, 30, 31, 32]. This makes the option to wire equivariance to the input order into the architecture desirable. Another issue is that RegretNet assumes a constant number of participants and items. This severely limits its practical applicability due to the inability to generalize to unseen input sizes and learn from data with heterogenous input sizes. Finally, the fully-connected layers used in RegretNet have limited expressivity and may not have the right inductive biases for auction design. As a remedy, we propose *RegretFormer*.

The architecture of RegretFormer is illustrated in Figure 1. The network takes the matrix of bids $B^{nm}$ as the input. First, we apply an exchangeable layer [25] to transform each bid into an initial vector of features that contains information about other bids. Then, we apply item-wise (i.e. row-wise) and participant-wise (i.e. column-wise) self-attention layers [60] to each feature vector corresponding to each bid. For a given bid, the outputs of the two self-attention layers are transformed into a single feature vector through a fully-connected layer. These self-attention transformations can be applied several times. Finally, by averaging over rows and columns we transform the output of self-attention layers into the allocation matrix $Z^{nm}$ and the payment vector $P^n$. Note that each layer shares the parameters across all bids to ensure that the network is insensitive to the order of bids and is applicable to different input sizes. The detailed definitions and order of all layers are provided in the Appendix.

Our architecture has several advantages. On the one hand, it leverages the expressivity of attention layers that help to achieve state-of-the-art performance in many diverse and complex problems, e.g. [60, 12, 63]. We empirically demonstrate the superiority of RegretFormer in Section 4. On the other hand, it maintains the equivariance and the invariance to the order of items and participants. The former forces the resulting mechanisms to be symmetric, which drastically reduces the solution search space. The latter allows RegretFormer to learn from data with inconstant input sizes, as well as to generalize to unseen settings. Note that RegretFormer can still learn asymmetric mechanisms by utilizing positional encoding, which we demonstrate empirically.

We note that a concurrent work by Duan et al. [14] also combines Transformers and RegretNet. However, their focus is on integrating contextual information of bidders and items into the framework, whereas we perform a wider and more accurate comparison of neural architectures.

## 3.2 Specifying regret budget

Both RegretNet and EquivariantNet optimize a mixture of two conflicting objectives (3), namely revenue maximization and regret minimization, and control their trade-off with hyperparameters like initial values and schedules of the Lagrange multipliers. There are two issues with this approach. The first issue is sensitivity. These hyperparameters have to be precisely tuned for each experiment, and as Rahme et al. [52] show, may massively degrade the performance if selected improperly. The second issue is uninterpretability. While increasing any of $\lambda_i, \rho, \rho_\Delta$ tightens the regret budget of the learned mechanism, the exact effect of such changes on resulting revenue and regret is unpredictable. Furthermore, the recipe for hyperparameter selection in new settings is unclear. Rahme et al. [52] propose a mixture of objectives that mitigates the first issue, but the second issue remains unresolved.

We propose an alternative perspective that mitigates both issues. Instead of balancing the two objectives, we maximize the revenue given a maximal regret budget $R_{max}$, which is pre-specified by the designer. This corresponds to a relaxed version of the constrained objective (1):

$$\min_{w} \quad -\frac{1}{|L|} \sum_{l \in L} \sum_{i \in N} p_i(v^l; w)$$
$$s.t. \quad \frac{1}{|L|} \sum_{l \in L} \sum_{i \in N} rgt_i(v^l; w) \leq R_{max} \tag{5}$$

Instead of the constrained objective (5), we optimize its dual by introducing the Lagrange multiplier $\gamma$ (note that $R_{max}$ does not depend on $w$ and thus can be dropped):

$$\mathcal{L}_{outer}(w) = -\sum_{i \in N} P_i + \gamma \sum_{i \in N} \widetilde{R}_i \tag{6}$$

Critically, unlike the original RegretNet we do not hand-select the Lagrange multiplier. Instead, $\gamma$ is dynamically updated to enforce the constraint satisfaction while exhausting all available regret

budget $R_{max}$. To this end, we employ dual gradient descent [4]. Specifically, we iterate between one gradient update of the network parameters $w$ to minimize (6) and one update of $\gamma$ according to:

$$\gamma \leftarrow \max\left(0, \gamma + \gamma_\Delta \left(\log(\sum_{i \in N} \widetilde{R}_i / \sum_{i \in N} P_i) - \log(R_{max})\right)\right) \tag{7}$$

where $\gamma_\Delta$ is the learning rate for the dual variable. For convenience, in (7) we normalize the regret estimate by revenue. This way, $R_{max} \in [0, 1]$ specifies the regret budget as a percentage of revenue. We also apply logarithms to both terms of the difference in (7) for faster convergence of $\gamma$.

Additionally, we apply a decreasing schedule to $R_{max}$. If the regret budget $R_{max}$ is too tight at the beginning of training, the network may fail to escape the local optima of low revenue and zero regret. To avoid that, we initialize $R_{max}$ at a higher value $R_{max}^{start}$ and exponentially anneal it to the desirable budget $R_{max}^{end}$ throughout the training. This leads the network to first find the solutions with high revenue and then tighten the regret. This results in the following update rule for the regret budget:

$$R_{max} \leftarrow \max\left(R_{max}^{end}, R_{max}^{mult} \cdot R_{max}\right) \tag{8}$$

where $R_{max}^{mult} < 1$ controls the speed of annealing of $R_{max}$ to $R_{max}^{end}$.

Another example of applying dual gradient descent to enforce a constraint can be found in [48].

The proposed approach has several advantages. On the one hand, it resolves the dichotomy of two conflicting objectives. While the regret budget needs to be explicitly set based on the designer's preferences, all other hyperparameters are then straightforward to tune by maximizing the revenue given the specified regret budget. This is unlike the original objective of RegretNet (3) where the regret budget is chosen implicitly through specifying uninterpretable loss-related hyperparameters like $\lambda_i$, $\rho$, $\rho_\Delta$. On the other hand, our approach is also significantly less sensitive to loss-related hyperparameters. We empirically demonstrate this by using the same hyperparameters related to the budget $R_{max}$ and its schedule in all our experiments, in contrast to the objective (3) that requires a uniquely tuned set of loss-related hyperparameters to perform well in a given setting [52].

## 4 Experiments

In this section, we empirically investigate our modifications. We compare our RegretFormer with RegretNet and EquivariantNet in settings with a constant number of participants and items used by Dütting et al. [17], as well as in novel settings with inconstant input sizes that we denote as multi-settings. All three networks are trained given the same regret budgets using our approach from Section 3.2. Additionally, we vary regret budgets and further investigate the differences between the architectures using novel validation procedures. In all experiments, the valuations of all participants are additive and are independently drawn for each item from the Uniform distribution: $v_i(j) \in \mathbb{U}[0, 1]$. Training details, hyperparameters, and additional results are reported in the Appendix.

### 4.1 Comparison of architectures under constant input sizes

The settings only differ in the number of participants $n$ and items $m$, so we denote them as $n$x$m$. We conduct experiments in five settings: {1x2, 2x2, 2x3, 2x5, 3x10}. The 1x2 setting is the celebrated Manelli-Vincent auction, the analytical solution for which is provided in [37]. The optimal revenue for this auction equals $0.55$. For the rest of the settings, the analytical solutions are unknown.

We compare our and existing neural architectures in the five settings given two different regret budgets $R_{max} \in \{10^{-3}, 10^{-4}\}$. We additionally include the classic VCG [61, 5, 22] and Myerson [42] mechanisms for comparison, both of which are DSIC and ex-post IR. The Myerson auctions are optimal for $m = 1$. We report its two extensions to multi-good settings: the 'item-wise' where each item is sold separately and the 'bundled' where all items are sold as a single bundle.

We report the results in Table 1. Additionally, we report the learning curves in the Appendix.

In all settings but 1x2 and given both regret budgets, RegretFormer outperforms both RegretNet and EquivariantNet in revenue. The performance gap is absent in the simplest setting but becomes

Table 1: Architecture comparison. Like in [16], revenue is summed over participants, and regret is averaged over participants. The highest revenue in a setting is highlighted in bold font. For brevity, we only report aggregated standard deviations: the average standard deviation of revenue is 0.006 for 1x2, 0.011 for 2x2, 0.009 for 2x3, 0.033 for 2x5, 0.019 for 3x10; the average standard deviation of regret is 0.00018 for $R_{max} = 10^{-3}$, 0.00003 for $R_{max} = 10^{-4}$.

| $R_{max}$ | setting | RegretNet | | EquivariantNet | | RegretFormer | |
| --- | --- | --- | --- | --- | --- | --- | --- |
| | | revenue | regret | revenue | regret | revenue | regret |
| $10^{-3}$ | 1x2 | 0.572 | 0.0007 | **0.586** | 0.00065 | 0.571 | 0.00075 |
| | 2x2 | 0.889 | 0.00055 | 0.878 | 0.0008 | **0.908** | 0.00054 |
| | 2x3 | 1.317 | 0.00102 | 1.365 | 0.00084 | **1.416** | 0.00089 |
| | 2x5 | 2.339 | 0.00142 | 2.437 | 0.00146 | **2.453** | 0.00102 |
| | 3x10 | 5.59 | 0.00204 | 5.744 | 0.00167 | **6.121** | 0.00179 |
| $10^{-4}$ | 1x2 | 0.551 | 0.00007 | 0.548 | 0.00013 | **0.556** | 0.00014 |
| | 2x2 | 0.825 | 0.00005 | 0.75 | 0.00005 | **0.861** | 0.00006 |
| | 2x3 | 1.249 | 0.00007 | 1.226 | 0.0001 | **1.327** | 0.00011 |
| | 2x5 | 2.121 | 0.00013 | 2.168 | 0.00017 | **2.339** | 0.00015 |
| | 3x10 | 5.02 | 0.00062 | 5.12 | 0.00025 | **5.745** | 0.00022 |

| $R_{max}$ | setting | VCG | | Myerson item-wise | | Myerson bundled | |
| --- | --- | --- | --- | --- | --- | --- | --- |
| | | revenue | regret | revenue | regret | revenue | regret |
| – | 1x2 | 0 | 0 | 0.5 | 0 | 0.544 | 0 |
| | 2x2 | 0.667 | 0 | 0.833 | 0 | 0.839 | 0 |
| | 2x3 | 1 | 0 | 1.25 | 0 | 1.278 | 0 |
| | 2x5 | 1.667 | 0 | 2.083 | 0 | 2.188 | 0 |
| | 3x10 | 5 | 0 | 5.312 | 0 | 5.003 | 0 |

Table 2: Ratio of the estimated regret to the regret budget. Our approach from Section 3.2 implies that this ratio should be close to 1 during training. Optimal misreports are estimated by minimizing (4) using 50 and 1000 gradient descent steps during training and validation, respectively.

| $R_{max}$ | setting | RegretNet | | EquivariantNet | | RegretFormer | |
| --- | --- | --- | --- | --- | --- | --- | --- |
| | | train | valid | train | valid | train | valid |
| $10^{-3}$ | 1x2 | 1.12 | 1.22 | 1.04 | 1.11 | 1.01 | 1.31 |
| | 2x2 | 0.97 | 1.24 | 1.41 | 1.82 | 0.89 | 1.19 |
| | 2x3 | 1.07 | 1.55 | 1.11 | 1.23 | 1.02 | 1.26 |
| | 2x5 | 0.94 | 1.21 | 1.11 | 1.2 | 0.8 | 0.83 |
| | 3x10 | 0.89 | 1.09 | 0.9 | 0.87 | 1.03 | 0.88 |
| $10^{-4}$ | 1x2 | 0.94 | 1.27 | 0.92 | 2.37 | 1.31 | 2.52 |
| | 2x2 | 0.95 | 1.94 | 1.73 | 1.33 | 0.93 | 1.39 |
| | 2x3 | 1.52 | 1.12 | 1.57 | 1.63 | 1.6 | 1.66 |
| | 2x5 | 1.04 | 1.23 | 1.02 | 1.57 | 0.95 | 1.28 |
| | 3x10 | 0.9 | 3.71 | 1.05 | 1.46 | 0.88 | 1.15 |

especially prominent in the larger settings. While permutation-equivariance of RegretFormer likely plays a role, it cannot fully explain the results since EquivarintNet also has this property. We, therefore, attribute the superiority of RegretFormer to the expressivity of attention layers.

Additionally, in Figure 2, we provide solutions found by RegretNet and RegretFormer in the 1x2 setting. Both networks find allocation probabilities that smoothly approximate the optimal solution.

In the bigger settings, EquivariantNet can also outperform RegretNet. This result is somewhat surprising since it was not observed by the authors of EquivariantNet, but it can simply be explained by the fact that they do not test the architecture in settings that are complex enough.

Given the smallest regret budget, both baselines sometimes underperform Myerson. While Regret-Former outperforms Myerson in revenue, its regret is still non-zero. This highlights a potential

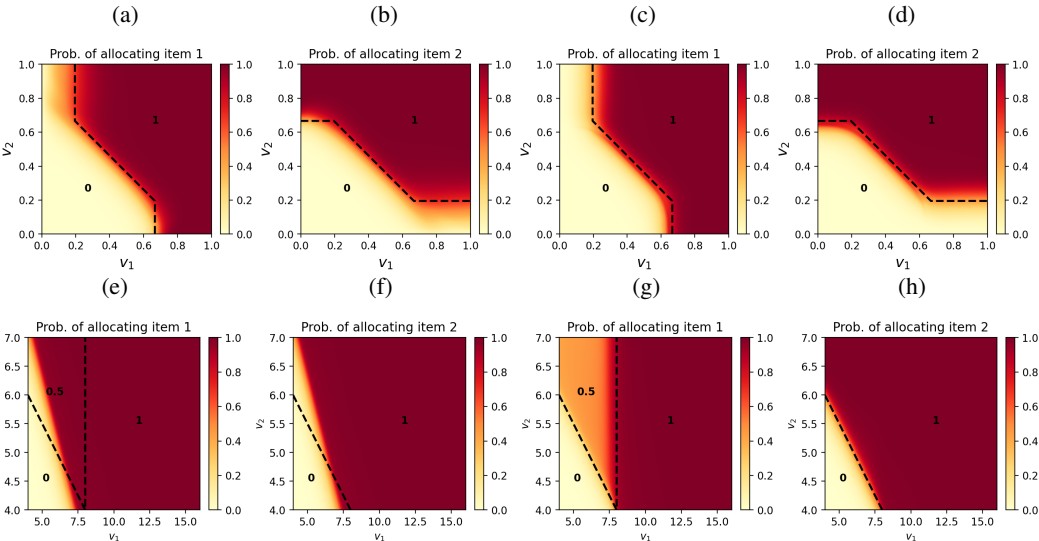

Figure 2: Allocation probabilities: (a, b) RegretNet in 1x2; (c, d) RegretFormer in 1x2; (e, f) RegretFormer without PE in the asymmetric setting; (g, h) RegretFormer with PE in the asymmetric setting. The optimal mechanisms are described by the regions separated by the dashed black lines, with the numbers in black denoting the optimal probability of allocation in the region.

problem of the regret-based approach: when any violations of DSIC are highly undesirable, better revenue might be achieved with classic mechanisms that are also guaranteed to be DSIC.

A potential downside of RegretFormer compared to the baselines is that it takes longer to train and requires more parameters to reach optimal performance. We elaborate on this in the Appendix.

## 4.2 Does regret budget function as indented?

The core feature of our approach from Section 3.2 is that the resulting regret should be uniquely and unambiguously specified through the regret budget. In particular, the total regret normalized by the total revenue should equal the regret budget. To verify this quantitatively, we present the ratios of the estimated normalized total regret to the regret budget in Table 2. A ratio higher than 1 means that the regret budget is exceeded, and vice versa. Below we summarize our findings.

During training, the ratios are indeed close to 1 for all three architectures, meaning that our approach functions as intended. Additionally, we report the ratios estimated during validation, which involves more gradient descent steps to find optimal misreports than during training, making the regret estimates more precise. Unsurprisingly, these are usually than the ratios estimated during training. For a tighter regret approximation and a better match with the budget, one could increase the number of the inner optimization steps during training, provided that a longer training time is acceptable.

## 4.3 Is the performance gap genuine?

In Table 1, we have observed that RegretFormer outperforms both baselines and have attributed its success to the expressivity of attention layers. However, there is an alternative explanation.

Recall that regret is approximated in the inner optimization (4) by repeatedly back-propagating through the whole neural network $w$. It follows that regret approximation depends on the neural architecture, including the number, the types, the size, and the order of layers. Due to the absence of a regret approximation procedure that is identical for all networks, there is no guarantee that similar regret estimates between different neural architectures correspond to actual similar regret values, i.e. for different architectures $w_1, w_2 : \tilde{R}(w_1) \approx \tilde{R}(w_2) \implies R(w_1) \approx R(w_2)$, where $R(w)$ is the expected regret and $\tilde{R}(w)$ is its approximation. In particular, we are concerned whether RegretFormer achieves higher revenues due to approximating the regret worse, rather than due to maximizing the revenue better. We design two procedures to test this hypothesis.

Table 3: Cross-misreport regret estimates. The highest regret for a network is highlighted with bold.

| setting | regret of | misreports of | | |
| --- | --- | --- | --- | --- |
| | | RegretNet | EquivariantNet | RegretFormer |
| 1x2 | RegretNet | **0.00079** | 0.00043 | 0.00074 |
| | EquivariantNet | 0.00102 | 0.00071 | **0.00129** |
| | RegretFormer | 0.00076 | 0.00046 | **0.00092** |
| 2x2 | RegretNet | **0.00050** | 0.00031 | 0.00024 |
| | EquivariantNet | 0.00021 | **0.00050** | 0.00022 |
| | RegretFormer | **0.00065** | 0.00056 | 0.00059 |
| 2x3 | RegretNet | **0.00116** | 0.00062 | 0.00044 |
| | EquivariantNet | 0.00020 | **0.00071** | 0.00028 |
| | RegretFormer | 0.00072 | 0.00087 | **0.00094** |
| 2x5 | RegretNet | **0.00149** | 0.00065 | 0.00060 |
| | EquivariantNet | 0.00075 | **0.00142** | 0.00071 |
| | RegretFormer | 0.00089 | **0.00120** | 0.00109 |
| 3x10 | RegretNet | **0.00198** | 0.00035 | 0.00012 |
| | EquivariantNet | 0.00035 | **0.00168** | 0.00014 |
| | RegretFormer | 0.00178 | 0.00178 | **0.00222** |

Table 4: Asymmetric setting from Daskalakis et al. [11], $R_{max} = 10^{-3}$

| Optimal | | RegretNet | | RegretFormer | | RegretFormer + PE | |
| --- | --- | --- | --- | --- | --- | --- | --- |
| revenue | regret | revenue | regret | revenue | regret | revenue | regret |
| 9.781 | 0 | 9.951 | 0.0072 | 9.967 | 0.0102 | 10.056 | 0.0099 |

The first procedure is denoted as 'cross-misreports'. It is based on estimating the regret of one network on the optimal misreports approximated by another network: $\widetilde{R}_i^{cross}(w_1, w_2) = \frac{1}{|B|} \sum_{l \in B} \widetilde{rgt}_i(v_i'^l(w_2), v^l; w_1)$. We apply this procedure to the trained networks in all five settings given $R_{max} = 10^{-3}$. If all networks approximate regret equally well, we expect the regret estimated on cross-misreports to not exceed the normally computed regret. However, if the regret estimates of RegretFormer were higher on the misreports estimated by RegretNet or EquivariantNet, this would point toward poor regret approximation by RegretFormer.

We report the results in Table 3. We do not find any evidence that RegretFormer underestimates the regret: in all settings, it finds misreports that produce regret either higher than or within a standard deviation from the misreports of the other two architectures.

The second procedure is based on network distillation. The results are similar to Table 3 in that RegretFormer does not appear to underestimate the regret, so we report them in the Appendix.

## 4.4 Learning asymmetric mechanisms using positional encoding

To handle asymmetric distributions, our architecture can be augmented with Positional Encoding (PE) - a common technique for transformers to incorporate information about the order of elements. To demonstrate this, we study a setting with one bidder, two items, and non-identically distributed valuations over items, where $v_1 \sim U[4, 16]$ and $v_2 \sim U[4, 7]$. The optimal mechanism is given by Daskalakis et al. [11]. We apply RegretNet, RegretFormer without PE, and RegretFormer with PE given $R_{max} = 10^{-3}$. As PE, we use a common technique that adds arbitrary numbers from [-1, 1] to the input (estimated as sine and cosine functions of position-dependent frequencies) proposed by Vaswani et al. [60]. Note that this PE has no learnable parameters.

The results are reported in Table 4. While RegretFormer with PE unsurprisingly performs on par with RegretNet, RegretFormer without PE learns a symmetric solution that is not much worse. In Figure 2, we provide illustrations of solutions found by the two versions of RegretFormer. These show that RegretFormer with PE closely approximates the allocation probabilities of the optimal mechanism,

Table 5: Multi-setting results. We report the highest regret identified with the cross-misreport procedure (see full cross-misreport results in the Appendix). (*) Note that EquivariantNet achieves unrealistically high revenue for a much higher regret.

| Setting | RegretNet | | EquivariantNet[(*)] | | RegretFormer | |
|---|---|---|---|---|---|---|
| | revenue | regret | revenue | regret | revenue | regret |
| average | 2.573 | 0.00352 | 2.889 | 0.00989 | 2.703 | 0.00391 |
| 2x3 | 1.386 | 0.00305 | 1.504 | 0.00554 | 1.432 | 0.00246 |
| 2x4 | 1.855 | 0.00341 | 2.070 | 0.00925 | 1.951 | 0.00317 |
| 2x5 | 2.339 | 0.00362 | 2.646 | 0.01270 | 2.471 | 0.00391 |
| 2x6 | 2.866 | 0.00425 | 3.226 | 0.01597 | 3.006 | 0.00439 |
| 2x7 | 3.346 | 0.00457 | 3.834 | 0.01951 | 3.553 | 0.00481 |
| 3x3 | 1.652 | 0.00322 | 1.795 | 0.00358 | 1.708 | 0.00251 |
| 3x4 | 2.217 | 0.00264 | 2.449 | 0.00508 | 2.312 | 0.00336 |
| 3x5 | 2.786 | 0.00277 | 3.108 | 0.00709 | 2.911 | 0.00421 |
| 3x6 | 3.362 | 0.00340 | 3.787 | 0.00916 | 3.521 | 0.00476 |
| 3x7 | 3.921 | 0.00430 | 4.467 | 0.01101 | 4.140 | 0.00553 |

whereas RegretFormer without PE finds a symmetric solution that resembles some middle ground between the optimal allocation probabilities for the two items.

### 4.5 Comparison of architectures in multi-settings

In practice, it may be desirable for a single network to be applicable to multiple input sizes, e.g. to save computation or due to limited data. To test our network in such cases, we introduce *multi-setting*. A multi-setting is a uniform mixture of constant-sized settings studied so far. In our experiments, we mix the following settings: {2x3, 2x4, 2x5, 2x6, 2x7, 3x3, 3x4, 3x5, 3x6, 3x7}. We set $R_{max} = 10^{-3}$. To adapt RegretNet to multi-settings, we fix the input size for the maximum possible number of items and participants in a multi-setting and pad unused valuations with zeros.

The results are presented in Table 5. Comparing RegretFormer with RegretNet, our architecture consistently achieves higher revenue for the same regret budget. While EquivariantNet achieves even higher revenue, after applying our cross-misreport validation procedure we find that EquivariantNet fails to minimize regret due to poorly approximating optimal misreports. Full results of cross-misreport validation, as well as out-of-setting experiments, are reported in the Appendix.

## 5 Conclusion

In this study, we achieve new state-of-the-art in the application of deep learning to optimal auction design. Our RegretFormer leverages recent advances in deep learning to unlock the full potential of regret-based optimization while enforcing the equivariance and the invariance to the order of participants and items. We test the effectiveness of RegretFormer in multiple experimental settings and find that our network consistently outperforms the existing analogues. In addition, we rethink the objective formulation of RegretNet. The resulting loss function disentangles balancing the revenue-regret trade-off and optimizing the performance, as well as reduces the burden of hyperparameter tuning. Finally, we suggest two validation procedures for comparing regret-based approaches that may find use in future studies.

## Acknowledgments

*All authors*. We sincerely thank Xeniya Adayeva for creating the illustration of RegretFormer 1. You can contact Xeniya at *xeniya.adayeva@gmail.com* regarding scientific illustrations.

*Dmitry Ivanov*. This research was supported in part through computational resources of HPC facilities at HSE University, Russian Federation. Support from the Basic Research Program of the National Research University Higher School of Economics is gratefully acknowledged.

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
