# OpenReview forum: "Optimal-er Auctions through Attention"
_NeurIPS.cc/2022/Conference — NeurIPS 2022 Accept_

### Official Review · Reviewer_M3wB · 2022-07-10

**Rating:** 8
**Confidence:** 4
**Soundness:** 4 excellent
**Presentation:** 4 excellent
**Contribution:** 4 excellent

**Summary:**

The use of deep learning for mechanism design is an expanding area. One important paper in the field, RegretNet, directly uses deep neural networks to approximate strategyproof and revenue-maximizing auctions. The current work builds on RegretNet in several ways. First, it uses an attention-based architecture -- this seems to work well, and has the key advantage of being permutation-equivariant (a desirable property, as it is known that permutation-equivariant mechanisms are optimal for symmetric auction design problems and often good in general). It also allows networks trained on one input size to generalize successfully to other input sizes. Second, the authors prevent a modified training algorithm that makes it easier to specify an acceptable small violation of the strategyproofness constraint, while reducing the complexity of tuning RegretNet's strange hyperparameters. The authors test their method and show performance that usually beats previous methods across a wide range of settings.

**Questions:**

- How might your architecture be modified to support unit-demand constraints (i.e. both rows and columns of the allocation matrix sum to <=1)? Is it as straightforward as just tacking the RegretNet approach on the end?

- How might your method be modified when equivariance is undesirable (e.g. for non-symmetric valuation distributions where you really do want to discriminate against one bidder)?

**Limitations:**

- The claim, mentioned as an aside, that the Myerson auction relies on having the full parametric valuation distribution seems not entirely correct to me, and I think this should be clarified. For one thing, Duetting et al. gives the "MyersonNet" architecture in their appendix; for another, there is a lot of work (cited with admirable thoroughness in the related work section) on sample complexity of revenue maximizing auctions where you can learn to approximate the right reserve price. I think methods like these would count as a "Myerson baseline" but work fine from samples.

- The authors are honest about where their work has limitations (e.g. low regret budget underperforming Myerson). I think a little bit more discussion and speculation about reasons the performance of the high/low regret budgets is warranted, although the NeurIPS page limit may prevent this.

**Strengths And Weaknesses:**

- Originality
    - It's a fairly natural idea to use transformers for auctions, because apparently people can use them for everything else too. There is concurrent work (CITransNet of Duan et al.) that also uses transformers and has some of the same advantages (generalization, equivariance).
    - In addition to the transformer idea, this paper presents some useful techniques, either explicitly claimed as contributions (e.g. the tweaked training objective) or implicitly (the methodology for making sure the performance gap is real).
- Significance:
    - I think this is a significant and valuable piece of work. It presents new state-of-the-art results for automated mechanism design, a new training procedure which is easier to work with, and is very methodologically careful in the way it evaluates performance.
- Quality
    - The quality is high, especially in the extremely thorough experiments. It's particularly nice that the authors take the time (section 4.2) to do experiments to make sure apparent outperformance is not just due to "gradient obfuscation" interfering with the misreport procedure, but is more likely due to actually finding a better mechanism.
    - The actual results in terms of performance don't seem like a complete slam dunk. It is true that in basically all cases, the RegretFormer *architecture* does better than the RegretNet/EquivariantNet architectures.
And for the medium 10^{-4} regret budget, things look great. But for the high regret budget, revenues seem oddly high across all networks. And for the low regret budget, revenues are depressingly low. So I am convinced that the RegretFormer architecture is better than previous alternatives, but have some reservations about the new training method (although it still seems to have advantages).
- Clarity
    - Everything is explained quite clearly.

---

> ### Author Response · Authors · 2022-08-02
> **Response to Reviewer M3wB**
>
> We thank the reviewer for their valuable feedback, as well as for spending time on reviewing our paper! We would like to answer the reviewer’s questions, as well as address some of their concerns.
>
> >How might your architecture be modified to support unit-demand constraints (i.e. both rows and columns of the allocation matrix sum to <=1)? Is it as straightforward as just tacking the RegretNet approach on the end?
>
> Yes, it is as straightforward, since both networks have identical outputs that can be identically modified, i.e. by taking minimum over the outputs of item-wise and participant-wise softmax applied to logits of the allocation network. As an alternative, both RegretNet and RegretFormer could be modified with Sinkhorn operator, i.e. by repeatedly taking softmax over rows and columns until approximate convergence to bistochastic matrix.
>
> >How might your method be modified when equivariance is undesirable (e.g. for non-symmetric valuation distributions where you really do want to discriminate against one bidder)?
>
> To handle asymmetric distributions, our architecture can be augmented with Positional Encoding (PE) - a common technique for transformers used to incorporate information about order of elements (e.g. order of words in a sentence). To demonstrate this application, we have performed an additional experiment that we describe as a separate comment and add as Section 3.5 to the revised Appendix. We find that PE indeed enables RegretFormer to learn optimal asymmetric auctions.
>
> >The claim, mentioned as an aside, that the Myerson auction relies on having the full parametric valuation distribution seems not entirely correct to me…
>
> We thank the reviewer for the correction. We will remove this claim from the text.

---

### Official Review · Reviewer_Yfdp · 2022-07-11

**Rating:** 6
**Confidence:** 3
**Soundness:** 2 fair
**Presentation:** 3 good
**Contribution:** 3 good

**Summary:**

This paper presents a new architecture for auction design, RegretFormer, which is based on the Transformer architecture. Unlike the original RegretNet, RegretFormer is invariant to the number of items and bidders and is permutation equivariant due to attention mechanisms. In addition, the authors relax the loss objective by introducing a regret budget $R_{max}$ to allow for smoother training and more robustness to hyperparameters. RegretFormer is shown to outperform baselines in both revenue and regret overall. Moreover, the authors confirm the validity of the regret estimates via a novel cross-misreports approach where the regret of one network is estimated on the optimal misreports approximated by another network. The results show that there is little evidence of over-estimate/bias in the regret estimates.

**Questions:**

- What is the size of your networks in terms of number of parameters compared to baselines? It would nice to see these number to ensure that the comparison between baselines is fair enough.
- How robust is RegretFormer to different hyperparameters compared to RegretNet and EquivariantNet?
- The generalization results in Table 3 show that RegretFormer outperforms EquivariantNet when trained on 3x10 setting but the same is not true for smaller settings like 1x2 and 2x2? Do you have intuition on why this is the case?
- How much does RegretFormer benefit from the modified loss objective? An small ablation study would be helpful here.

**Limitations:**

While the results are positive, I found the paper to be lacking in terms of analysis of the results to better understand the advantages/disadvantages of RegretFormer. For example, how sample efficient is RegretFormer compared to baselines?

**Strengths And Weaknesses:**

### Strengths:
- The RegretFormer architecture offers desirable properties of invariance and permutation equivariance due to attention.
- The authors propose a smoother loss function that allows for pre-specification of a regret budget.
- The cross-misreport experiments offer a nice approach to validate the regret estimates which may be heavily biased by the trained network.

### Weakness:
- The authors claim that there modified loss function is more robust to hyperparameters. However, there are no experiments to prove this. It would be nice to see the performance under different hyperparameters compared to baselines.
- While RegretFormer is well justified for this problem, the architecture itself is a straightforward application of the transformer architecture.

---

> ### Author Response · Authors · 2022-08-02
> **Response to Reviewer Yfdp**
>
> We thank the reviewer for their valuable feedback, as well as for spending time on reviewing our paper! We would like to answer the reviewer’s questions, as well as address some of their concerns.
>
> >The authors claim that there modified loss function is more robust to hyperparameters. However, there are no experiments to prove this. It would be nice to see the performance under different hyperparameters compared to baselines… How robust is RegretFormer to different hyperparameters compared to RegretNet and EquivariantNet?
>
> While we do not perform explicit additional experiments to verify robustness of our loss function, implicitly we do verify it in the fact that **we use the same loss-related hyperparameters for all architectures in all experiments** (i.e. regret budget $R_{max}$, its schedule parameters, and learning rate of Lagrange multiplier $\gamma_\Delta$). Their values are specified in Section 2 of the Appendix. This is unlike the experiments in the RegretNet paper where each setting requires a unique set of loss-related hyperparameters. Furthermore, Rahme et al. (2021) [44] show that applying loss-related hyperparameters of the original RegretNet objective tuned for one setting to another setting may result in drastic performance losses (Table 1 in their paper). Obviously, this is not the case for our loss function since networks perform well despite using the same loss-related hyperparameters in all settings. To be absolutely clear, we do not discuss or claim robustness to any other hyperparameters, e.g. depth and width of neural network or learning rate. Also, for clarity we stress that all three architectures (RegretNet, EquivariantNet, RegretFormer) are trained on our loss function proposed in Section 3.2. In our answer to Reviewer VVEG, we discuss in more detail how this choice results in fair comparison of architectures and even potentially improves performance of baselines.
>
> >How much does RegretFormer benefit from the modified loss objective? A small ablation study would be helpful here.
>
> The advantages of our loss function are qualitative rather than quantitative (robustness to loss-related hyperparameters and interpretability). This makes measuring the magnitude of these advantages difficult. Performance-wise, both loss functions can be applied to achieve similar revenue and regret. However, the original loss requires meticulous hyperparameter tuning (both to maximize performance and to control revenue-regret trade-off) whereas our loss does not (as discussed above).
>
> >What is the size of your networks in terms of number of parameters compared to baselines? It would nice to see these number to ensure that the comparison between baselines is fair enough.
>
> Please find the number of parameters of all networks, as well as a related ablation experiment, either in a separate comment denoted as ‘Ablation’ or in Section 3.4.2 of the revised Appendix.
>
> >The generalization results in Table 3 show that RegretFormer outperforms EquivariantNet when trained on 3x10 setting but the same is not true for smaller settings like 1x2 and 2x2? Do you have intuition on why this is the case?
>
> We believe this mirrors the results of the in-setting experiments (Table 1) where RegretFormer outperfors baselines only in the bigger settings while performing on par with baselines in smaller settings. Likewise, in the out-of-setting experiments RegretFormer trained on 3x10 setting learns better mechanisms than the baseline (and thus generalizes better), whereas this is not the case in smaller settings.
>
> >how sample efficient is RegretFormer compared to baselines?
>
> We train all architectures on the dataset of the same size and for the same number of iterations (these details are reported in Section 2 of the Appendix). The learning curves are illustrated in the end of the Appendix. From these curves we can derive that RegretFormer is as sample efficient as baselines. However, note that the curves are of the decreasing shape due to regret budget scheduling described in Section 1.2 of the Appendix. Further analysis without schedules is required to conclude if any of the architectures converges to the final performance faster (or if any of the architectures can learn from fewer data).

---

> > ### Comment · Reviewer_Yfdp · 2022-08-07
> > **Response to Authors**
> >
> > I would like to thank the reviewers for their detailed reply. Most of my concerns have been addressed. Though this work is not significantly novel, the analysis is thorough and the use of transformers is well motivated. I will keep my score the same.

---

> > > ### Author Response · Authors · 2022-08-08
> > > **Response to Follow-up**
> > >
> > > Thank you for the follow-up, and thank you again for reviewing our paper!

---

### Official Review · Reviewer_u7uQ · 2022-07-11

**Rating:** 4
**Confidence:** 5
**Soundness:** 3 good
**Presentation:** 4 excellent
**Contribution:** 2 fair

**Summary:**

This paper modifies the RegreNet architecture and training in the following ways -
(1) it replaces the neural network architecture with a transformer
(2) it replaces the augmented lagrangian optimization with a new loss formulation (rather than enforcing IC violation to be strictly 0, it only requires them to be at some threshold $R_{max}$

**Questions:**

(1) Is Eqn (6) missing a term perhaps something like $\max (\tilde{R} - R_{max}, 0)$ ?
(2) There are other symmetric distributions where this architecture can be tested - single bidder but from the uniform interval [c, c+1], uniform triangle settings/correlated valuations, and settings with unit-demand valuations where we know the optimal solution analytically.
(3) How is the regret computed at test time? How many samples of misreports are initialized and how many gradient descent steps to compute them? What is the size of the test set?
(4) Can you visualize the solution for the 1x2 setting? The optimal auction is given by Maneli and Vincent which has 4 deterministic menu options. This achieves a revenue of $0.55$. Selling it as a grand bundle has revenue that's slightly less than $0.55$ My intuition is that the setting when $R_{max} = 1e-5$, the latter is recovered based on the revenue values reported. If this is the case, perhaps the proposed optimization algorithm here is sensitive to $\gamma$ initialization as well.
(5) Would be possible to use tools from [Certifying Strategyproof Auction Networks](https://arxiv.org/abs/2006.08742) to compute regret more accurately?

**Limitations:**

Discussed above!

**Strengths And Weaknesses:**

**Strengths**:
The paper aims to tackle problems that are significant and relevant and are in the right direction - it would be nice to use optimization methods that were less sensitive to the hyperparameters than the augmented Lagrangian optimization method.

**Weaknesses**:
(1) The contribution is not significantly novel. The use of transformers was already proposed by [54]
(2) The transformer architecture has limitations - it can't handle asymmetric distributions.
(3) Eqn (6) doesn't seem to be a proper relaxation of the optimization problem in (5).
(4) The evaluation is limited. The authors only consider uniform distribution on the interval
(5) Missing details on regret computation at test time
(6) I'm not convinced that the cross-misreport testing rules out the possibility of regret being accurate. The loss landscapes are very different for different neural network architectures. While low regret on cross misreports is a necessary condition, it isn't a sufficient one.

---

> ### Author Response · Authors · 2022-08-02
> **Response to Reviewer u7uQ**
>
> We thank the reviewer for their valuable feedback, as well as for spending time on reviewing our paper! We would like to answer the reviewer’s questions, as well as address some of their concerns.
>
> >(1) The contribution is not significantly novel. The use of transformers was already proposed by [54].
>
> Could the reviewer clarify the citation? The citation [54] from our paper applies transformers to a computer vision task unrelated to auctions.
>
> As mentioned by Reviewer M3wB, there is a concurrent work (CITransNet of Duan et al.) that also applies Transformers to optimal auction design. There are, however, multiple differences between the two papers. Whereas they focus on integration of additional context, we focus on the revenue gains granted by our architecture, to which end we design novel validation procedures (cross-misreports and distillation) and novel experimental settings (multi-settings). On top of that, our paper presents an independent contribution of novel loss function. We believe these differences make the two papers complementary. We will clarify this connection in the text.
>
> >(2) The transformer architecture can't handle asymmetric distributions.
>
> To handle asymmetric distributions, our architecture can be augmented with Positional Encoding (PE). To demonstrate this, we have performed an additional experiment that we describe as a separate comment and add as Section 3.5 to the revised Appendix. We find that PE indeed enables RegretFormer to learn optimal asymmetric mechanisms.
>
> >Q(2) There are other symmetric distributions where this architecture can be tested - single bidder but from the uniform interval [c, c+1], uniform triangle settings/correlated valuations, and settings with unit-demand valuations.
>
> We would like to note that more experiments are provided in the Appendix, including novel “multi-settings” where the number of items and participants is not constant and which were not studied in the literature before. While shifting the uniform interval and making valuations correlated are also important settings, we do not expect any different conclusions from these experiments since these settings are not more difficult than the ones we study. Regarding unit-demand valuations, our architecture can be adapted to this setting the exact same way as RegretNet.
>
> >(6) I'm not convinced that the cross-misreport testing rules out the possibility of regret being accurate.
>
> We would like to note that in the Appendix (Table 5) we demonstrate a successful application of the cross-misreport validation procedure to identify a fake performance gap of EquivariantNet over our architecture. Additionally, we investigate a potentially more precise distillation validation procedure in Section 3.2 of the Appendix.
>
> >Q(1) Is Eqn (6) missing a term perhaps something like $\max(R - R_{max}, 0)$?
>
> No. $R_{max}$ is a constant that does not depend on the network’s parameters and therefore does not contribute to the gradient. The constraint is instead enforced through the precise fit of $\gamma$ (equation 7). To be clear, the regret and its multipier $\gamma$ are always non-negative in equation 6.
>
> Adding relu to equation 6 is likely to have detrimental effect: the moment the constraint on regret budget is satisfied, the regret term in the loss function becomes null and the mechanism is only focused on maximising revenue. This results in an objective that can never satisfy the desired constraint, no matter how high $\gamma$ is.
>
> >Q(3) How is the regret computed at test time? How many samples of misreports are initialised and how many gradient descent steps to compute them? What is the size of the test set?
>
> The requested details are already provided in Section 2 of the Appendix, lines 72-75:
>
> “The validation dataset consists of 4096 profiles divided into 128 batches of 32 profiles. The number of inner optimization steps per one outer update equals 25 during training and 1000 during validation.”
>
> Additionally, the learning curves and min-max intervals over three random seeds are illustrated in the end of the Appendix for all experiments from Table 1 in the main  text (these intervals are quite narrow).
>
> >Q(4) Can you visualize the solution for the 1x2 setting?
>
> We provide the requested illustrations in the revised Appendix (Section 3.6). These show that RegretFormer learns allocations probabilities similar to optimal for all three regret budgets.
>
> >Q(5) Would it be possible to use tools from Certifying Strategyproof Auction Networks to compute regret more accurately?
>
> We thank the reviewer for bringing this paper to our attention and will cite it in the future. Their certification method could potentially complement our distillation and cross-misreport validation procedures. However, we want to note that increasing the number of items substantially slows down certification (Table 2 in their paper) and thus is not applicable in settings with 5 to 10 items that we study.

---

> > ### Comment · Reviewer_u7uQ · 2022-08-08
> > **Follow-up**
> >
> > - Yes, I meant the CITransNet of Duan et al. Sorry for the confusion.
> > - Thanks for the thorough follow-up and clarifications!
> >
> > I just have one additional comment
> > - RegretNet is evaluated on 10000 samples (2.5x of what is reported in this paper). It would be nice if the authors could report results on sample sets of similar sizes.

---

> > > ### Author Response · Authors · 2022-08-08
> > > **Response to Follow-up**
> > >
> > > Thank you for the follow-up!
> > >
> > > Regarding the validation set, we choose a smaller sample size because our GPUs could not fit more data in a single batch during validation, whereas increasing the number of batches would significantly slow down the computation. However, since the confidence intervals are quite narrow (judging by the figures in the Appendix), as well as given the consistency of our architecture outperforming the baselines, we believe our evaluation protocol is adequate even with smaller sample size.
> > >
> > > Also, given that we have addressed most of your concerns, we would greatly appreciate a clarification on what could be further improved in our paper so you could consider recommending its acceptance.
> > >
> > > Thank you again for reviewing our paper!

---

> > > > ### Comment · Reviewer_u7uQ · 2022-08-09
> > > > **Maintaining my scores**
> > > >
> > > > I'm retaining my scores for the following reason:
> > > > - While the application is different, the core contribution is the same - applying a transformer-based architecture for auction design.
> > > >  - I do not find the loss function to be novel. Given that $R_{max}$, the new objective is just a trade of loss ( setting $\rho = 0$) in the original objective. The change now comes from how $\gamma$ is updated. By choosing an appropriate update step and frequency of updates, I believe the augmented lagrangian optimizer can emulate this. (Note that this would involve fixing some parameters throughout for all the settings so the number of hyperparameters that needs to be tuned also reduces to 1)
> > > > - The analytical solution for the $1\times2$ setting (a single buyer with additive valuation over two items drawn from U[0,1])  is around 0.55. Empirical methods report a revenue higher than this (this is because these mechanisms are not IC). Higher differences between the revenue of the empirical method vs. the analytical optimal are only indicative of the fact the evaluation is not accurate (either due to not having enough sample sizes or not evaluating regret accurately) This does not imply that the method/approach is *optimal-er*.
> > > > - The misreport analysis in both the main paper and the follow-up is a necessary condition for regret being accurate but not sufficient. Perhaps this paper can benefit a little bit from rewriting this section with more clarity.

---

> > > > > ### Author Response · Authors · 2022-08-10
> > > > > **Thank you for the clarification**
> > > > >
> > > > > Thank you for the clarification, we appreciate it!

---

### Official Review · Reviewer_VVEG · 2022-07-12

**Rating:** 6
**Confidence:** 3
**Soundness:** 3 good
**Presentation:** 3 good
**Contribution:** 3 good

**Summary:**

This paper introduces a transformer-based network architecture called RegretFormer to address the problem of optimal auction design. RegretFormer differs from prior optimization techniques that frame the problem as a mini-max between regret and revenue. Instead, RegretFormer defines a fixed regret budget and maximizes revenue. Maximum regret is used as a single interpretable hyperparameter to tune the auction design. The transformer architecture allows for variable sized inputs, and attention is used to enforce equivariance. RegretFormer only studies symmetric auctions, leaving asymmetric auctions (with the use of positional encoding) as future work. Different from prior works (c.f. RegretNet and EquivarientNet), RegretFormer uses a single network to predict both allocations and payments. Lastly, this paper proposes two supplemental evaluation protocols, namely cross-misreport regret estimates and network distillation to verify that the performance improvement over prior works is genuine.


**Questions:**

1. How long does it take to train RegretFormer when compared to RegretNet and EquivariantNet. How does this scale according to the size of the auction?
2. Since the distribution of bids are different depending on the size of the auction, it isn't clear why we would expect "zero-shot" out-of-setting generalization by default. Is it the case that the RegretFormer architecture allows us to fine tune for new settings with fewer examples?
3. Does evaluating with the proposed "fair" protocol defined above change any conclusions?


**Limitations:**

Authors highlight that although modulating the maximum regret parameter should proportionally change revenue, this sometimes does not hold. Since the interpretable maximum regret parameter is a key selling point of this work, it would be useful to implement the proposed solution (L247) to show that this tighter revenue/regret ratio holds. Authors also present general limitations of a regret-based approach to auction design, highlighting that under the smallest regret budget, regret is still non-zero.
It would be helpful to make limitations of this method an explicit subsection. The current writeup (L250) reads as "all methods have limitations, but RegretFormer does better", which isn't particularly helpful for understanding where RegretFormer fails.


**Strengths And Weaknesses:**

This paper is generally well written and clear. The main takeaways are clear and the authors appropriately juxtapose their work in the context of prior work (c.f. RegretNet and EquivariantNet) and make it clear how their work differs. The authors convincingly show that RegretFormer performs better than both RegretNet and EquivariantNet. Moreover, the authors ask the right question - is the performance gap genuine? Authors devise protocols for addressing this question.

The cross-misreport regret estimate, although well motivated, is not convincing because there is no empirical proof to suggest how much an adversarially constructed example would underestimate regret. Without this point of reference, it's difficult to interpret the results presented in Table 3 (given variance between training runs). However, the network distillation seems to have the opposite problem, the result is convincing, but since RegretFormer is able to outperform EquivariantNet and RegretNet, the motivation that network architecture of RegretFormer impairs its ability to approximate optimal misreports isn't a major concern. To be clear, these attempts at deeply understanding if the performance improvement by RegretFormer is genuine are appreciated.

One concern about evaluation is w.r.t setting a maximum regret for comparison (Table 1). From the main paper, it's not clear how maximum regret is enforced for RegretNet and  EquivariantNet since these are optimized as a mini-max of regret and revenue. According to Appendix 1.2, it seems like exponential annealing is used to achieve this constraint. However, since this is not how either RegretNet or EquivariantNet are normally trained and (L182) both models are sensitive to the hyperparameters and training protocol, is the comparison with RegretFormer at a fixed maximum regret fair? Here's a protocol that may address this concern: First, train RegretNet using the standard unconstrained (w.r.t maximum regret) optimization. Given the maximum regret of the trained RegretNet, set this is the maximum regret of RegretFormer and train the model. Now the comparison is fair. This protocol can be naturally extended to EquivariantNet. Ideally, it might be the case that RegretFormer still outperforms prior methods.

---

> ### Author Response · Authors · 2022-08-02
> **Response to Reviewer VVEG**
>
> We thank the reviewer for their valuable feedback, as well as for spending time on reviewing our paper! We would like to answer the reviewer’s questions, as well as address some of their concerns.
>
> > One concern about evaluation is w.r.t setting a maximum regret for comparison (Table 1)... since this is not how either RegretNet or EquivariantNet are normally trained and (L182) both models are sensitive to the hyperparameters and training protocol, is the comparison with RegretFormer at a fixed maximum regret fair?.. Does evaluating with the proposed "fair" protocol defined above change any conclusions?
>
> To avoid any possibility of misunderstanding, we would like to explicitly state that **all three architectures (RegretNet, EquivariantNet, RegretFormer) are trained identically using the novel loss function proposed in Section 3.2**. We apologise if this wasn’t clear from the paper. This is done exactly to ensure fair comparison. In our experiments, we ask: given the same regret budget, how much revenue can different architectures achieve? Notably, this procedure makes the two-step protocol proposed by the reviewer unnecessary.
>
> To ensure that the change of loss function does not impair the performance of RegretNet, we can cross-reference our results in Table 1 with the results reported in the original RegretNet paper (arxiv version): settings 1x2 (their Figure 5, setting A), 2x2 (their Figure 16, setting M), 2x3 (their Figure 10), 3x10 (their Figure 9, setting D). The comparison is summarized below.
>
> \\begin{array}{lllllll}
> \\hline
>  & \text{RegretNet} &  & \text{RegretNet} &  & \text{RegretNet} &  \\\\
>  & R_{max} = 10^{-3} \text{(our)} &  & R_{max} = 10^{-4} \text{(our)} &  & \text{Dutting et al. 2020} &  \\\\
> \\hline
> \text{setting} & \text{revenue} & \text{regret} & \text{revenue} & \text{regret} & \text{revenue} & \text{regret} \\\\
> 1x2 & 0.579 & 0.00123 & 0.55 & 0.00017 & 0.554 & \textless 0.001 \\\\
> 2x2 & 0.888 & 0.00066 & 0.849 & 0.00009 & 0.878 & \textless 0.001 \\\\
> 2x3 & 1.352 & 0.00102 & 1.265 & 0.00012 & 1.291 & \textless 0.001 \\\\
> 3x10 & 5.746 & 0.00292 & 5.016 & 0.00062 & 5.541 & \textless 0.002 \\\\
> \\hline
> \\end{array}
>
>
> Evidently, in our experiments RegretNet performs either on par with or better than in the original paper for similar regret levels, which is likely a consequence of our loss being less sensitive to hyperparameters (RegretNet could be undertuned in the original paper while not requiring such meticulous tuning in our experiments). This leads us to believe that if we implemented the protocol proposed by the reviewer, the performance gap of RegretFormer over RegretNet could either stay the same or become even more prominent.
>
> > The cross-misreport regret estimate, although well motivated, is not convincing because there is no empirical proof to suggest how much an adversarially constructed example would underestimate regret.
>
> We would like to note that in the Appendix (Table 5) we demonstrate a successful application of the cross-misreport validation procedure to identify a fake performance gap of EquivariantNet over our architecture.
>
> > How long does it take to train RegretFormer when compared to RegretNet and EquivariantNet. How does this scale according to the size of the auction?
>
> Below we report the average wall-clock time in hours. It takes longer to train RegretFormer than the baselines for two reasons. First, to perform optimally RegretFormer requires a bigger network with more parameters than baselines, especially in the bigger settings (for more details, please see a related ablation study provided as a separate comment and in Section 3.4.2 of the revised Appendix). Second, the attention layers have quadratic O(n^2) complexity (in our case, n is the number of items or participants). If training time is an issue, the attention layers can be replaced with one of multiple modifications that have O(n logn) or O(n) complexity.
>
>
> \\begin{array}{llll}
> \\hline
> \\text{setting} & \\text{RegretNet} & \\text{EquivariantNet} & \\text{RegretFormer} \\\\
> \\hline
> \\text{1x2} & 2.5 & 5.2 & 12.5 \\\\
> \\text{2x2} & 2.5 & 5.3 & 12.0 \\\\
> \\text{2x3} & 2.5 & 8.4 & 11.0 \\\\
> \\text{2x5} & 3.1 & 9.8 & 28.7 \\\\
> \\text{3x10} & 4.7 & 11.2 & 82.1 \\\\
> \\hline
> \\end{array}
>
>
> > Is it the case that the RegretFormer architecture allows us to fine tune for new settings with fewer examples?
>
> We thank the reviewer for the suggestion. This is an interesting experiment that we leave as future work.

---

> > ### Comment · Reviewer_VVEG · 2022-08-07
> > **Response to Authors**
> >
> > The authors have sufficiently answered my concerns and provided extensive evaluation of their proposed method. Though there are some concerns about novelty due to similar concurrent work using Transformers for Auction Design, the proposed framework RegretFormer is clearly described and rigorously evaluated. I am upgrading my score in light of the author's rebuttal. I request that the authors make their code public to facilitate future work in Auction Design using Deep Learning.

---

> > > ### Author Response · Authors · 2022-08-08
> > > **Response to Follow-up**
> > >
> > > Thank you so much for increasing your score!
> > >
> > > We will absolutely share our code once the paper is accepted. Given that it's written in PyTorch, it should complement nicely the code that accompanies RegretNet written in TensorFlow.
> > >
> > > Thank you again for reviewing our paper!

---

### Author Response · Authors · 2022-08-02
**Additional Experiment: Learning Asymmetric Mechanisms using Positional Encoding**

To handle asymmetric distributions, our architecture can be augmented with Positional Encoding (PE) - a common technique for transformers used to incorporate information about order of elements (e.g. order of words in a sentence). To demonstrate this application, we have performed an additional experiment.

We study a setting with one bidder, two items, and non-identically distributed valuations over items, where $v_1 \sim U[4, 16]$ and $v_2 \sim U[4,7]$. The optimal mechanism is given by Daskalakis et al. (2017) and provides the revenue of 9.781. This setting is also studied in the RegretNet paper (setting J in their arxiv version). In this setting, we apply RegretNet, RegretFormer without PE, and RegretFormer with PE. We set $R_{max} = 10^{-4}$. All hyperparameters are the same as for the 1x2 setting from the main text (the values are reported in Section 2 of the Appendix). The metrics are averaged over three seeds. As PE, we use the most common technique that adds arbitrary numbers from [-1, 1] to the input (estimated as sine and cosine functions of position-dependent frequencies) proposed by Vaswani et al. (2017). Note that this PE has no learnable parameters. The results are reported in the table below.

$
\\begin{array}{llllll}
\\hline
\text{RegretNet} & & \text{RegretFormer} & & \text{RegretFormer + PE} & \\\\
\\hline
\text{revenue} & \text{regret} & \text{revenue} & \text{regret} & \text{revenue} & \text{regret} \\\\
9.671 & 0.00077 & 9.605 & 0.00132 & 9.739 & 0.00079 \\\\
\\hline
\\end{array}
$

It is clear that RegretFormer + PE can approximate optimal mechanism in this asymmetric setting as well as RegretNet. What is more, RegretFormer without PE learns a symmetric solution that is not much worse. In the revised Appendix (Section 3.5), we provide illustrations of solutions found by the two versions of RegretFormer. These show that RegretFormer + PE (like RegretNet) very closely approximates the allocation probabilities of the optimal mechanism for both items, whereas RegretFormer without PE finds a symmetric solution that resembles some middle ground between the optimal allocation probabilities for the two items.

---

### Author Response · Authors · 2022-08-02
**Ablation: Varying Sizes of Networks**

$
\\begin{array}{llll}
\\hline
\\text{setting} & \\text{RegretNet} & \\text{EquivariantNet} & \\text{RegretFormer} \\\\
\\hline
\\text{1x2}     & 21305     & 4546           & 12705        \\\\
\\text{2x2}     & 22008     & 4546           & 49985        \\\\
\\text{2x3}     & 22711     & 12802          & 49985        \\\\
\\text{2x5}     & 84717     & 16930          & 362753       \\\\
\\text{3x10}    & 91343     & 16930          & 362753       \\\\
\\hline
\\end{array}
$

The table above presents the sizes of the three architectures in each setting from Table 1, measured in number of parameters. In all settings but 1x2, RegretFormer is larger than both baselines. To investigate the robustness of our results to the network sizes, we have performed an additional ablation study where we downscale RegretFormer to the sizes of the baselines and, vice versa, upscale the baselines to the size of RegretFormer. We do this by changing both depth and width of the neural networks. We perform this ablation in the 3x10 setting where the performance gap between the architectures is the most prominent. We set $R_{max} = 10^{-3}$. The results are reported in tables below.

$
\\begin{array}{llllll}
\\hline
\text{RegretFormer} & & \text{RegretFormer} & & \text{RegretFormer} \\\\
\text{362753 weights} & & \text{91265 weights} & & \text{20897 weights} \\\\
\\hline
\text{revenue} & \text{regret} & \text{revenue} & \text{regret} &\text{revenue} & \text{regret} \\\\
6.157 & 0.00332 & 6.117 & 0.00208 & 5.778 & 0.00255 \\\\
\\hline
\\end{array}
$

$
\\begin{array}{llllllll}
\\hline
\text{RegretNet} & & \text{RegretNet} & & \text{EquivariantNet} & & \text{EquivariantNet} \\\\
\text{91343 weights} & & \text{382893 weights} & & \text{16930 weights} & & \text{264322 weights} \\\\
\\hline
\text{revenue} & \text{regret} &\text{revenue} & \text{regret} &\text{revenue} & \text{regret} &\text{revenue} & \text{regret}  \\\\
5.746 & 0.00292 & 5.72 & 0.0023 & 6.057 & 0.00305 & 6.014 & 0.00323 \\\\
\\hline
\\end{array}
$

When downscaling RegretFormer, its performance suffers. When its size is reduced roughly to the size of RegretNet, it performs slightly worse but still outperforms the baselines. More extreme downscaling significantly hinders the performance. However, when upscaling the baselines, their performance suffers as well. From these results, we can conclude that RegretFormer requires a larger network than the baselines to perform optimally, but has a higher performance ceiling. This conclusion begs a question whether RegretFormer can benefit from further scaling, which could be especially valuable when coupled with low regret budgets. We leave answering this question as future work.

---

### Meta-Review · Area_Chair_ekqL · 2022-08-30

**Recommendation:** Accept
**Confidence:** Certain

**Metareview:**

This paper builds on the nascent optimal auction design via deep learning literature.  It modifies the at-this-point standard approach, RegretNet, to this setting from Duetting et al (ICML-19, but on arXiv earlier) by incorporating a transformer-based architecture into, roughly, the same training scenario, albeit with a different loss function that is more appropriate for this setting.  That loss function is a nice add-on in that RegretNet and kin are notoriously hard to train, and this seems to help.  From an empirical point of view alone, I believe this paper is worth accepting because it pushes the boundary on a tricky empirical problem, both in terms of ease of training but also in terms of (in many cases) the quality of the final network itself, measured in revenue / IC violation.  I agree with some of the negative sentiment from Reviewer u7uQ, though, w.r.t. over-claims of "optimal-er" and poor comparison against analytic baselines; this should be changed in a final/future version of the work.

**Award:**

No

---

### Decision · Program_Chairs · 2022-09-14

Accept